# Decrease in Mycophenolate Mofetil Plasma Concentration in the Presence of Antibiotics: A Case Report in a Cystic Fibrosis Patient with Lung Transplant

**DOI:** 10.3390/ijms25042358

**Published:** 2024-02-17

**Authors:** Giuliano Ponis, Giuliana Decorti, Egidio Barbi, Gabriele Stocco, Massimo Maschio

**Affiliations:** 1Institute for Maternal and Child Health, IRCCS “Burlo Garofolo”, 34137 Trieste, Italy; giuliano.ponis@burlo.trieste.it (G.P.); egidio.barbi@burlo.trieste.it (E.B.); massimo.maschio@burlo.trieste.it (M.M.); 2Department of Medicine, Surgery and Health Sciences, University of Trieste, 34149 Trieste, Italy; decorti@units.it

**Keywords:** amikacin, antibiotic therapy, drug interactions, immunosuppression, therapeutic drug monitoring, meropenem, mycophenolate mofetil

## Abstract

Immunosuppression management in transplant recipients is a critical component of pharmacotherapy. This becomes particularly crucial when patients are exposed to multiple medications that may lead to pharmacological interactions, potentially compromising the effectiveness of immunosuppression. We present the case of a 46-year-old patient diagnosed with cystic fibrosis in childhood at our hospital, who underwent bilateral lung transplantation and is undergoing immunosuppressive therapy. The patient was hospitalized due to an acute pulmonary exacerbation. During the hospitalization, the patient was administered various classes of antibiotics while continuing the standard antirejection regimen of everolimus and mycophenolate. Plasma concentrations of immunosuppressants, measured after antibiotic therapy, revealed significantly lower levels than the therapeutic thresholds, providing the basis for formulating the hypothesis of a drug–drug interaction phenomenon. This hypothesis is supported by the rationale of antibiotic-induced disruption of the intestinal flora, which directly affects the kinetics of mycophenolate. These levels increased after discontinuation of the antimicrobials. Patients with CF undergoing lung transplantation, especially prone to pulmonary infections due to their medical condition, considering the enterohepatic circulation of mycophenolate mediated by intestinal bacteria, necessitate routine monitoring of mycophenolate concentrations during and immediately following the cessation of antibiotic therapies, that could potentially result in insufficient immunosuppression.

## 1. Background

Cystic fibrosis (CF) is an autosomal recessive genetic disease common among the Caucasian population. It is caused by mutations involving the *CFTR* (Cystic Fibrosis Transmembrane Conductance Regulator) gene, which encodes for a glycoprotein that is crucial for chloride and bicarbonate ion transport. The mutation can impair CFTR expression or reduce its activity. Individuals affected by this condition secrete thick mucus in the lungs, obstructing normal mucociliary clearance and facilitating bacterial infections [1]. Despite the recent advancements introduced by recently approved “caftor” drugs (elexacaftor, ivacaftor, lumacaftor, and tezacaftor), there is currently no definitive cure for the disease. However, several therapies, including antibiotic therapy to control lung infections, can slow its progression. Lung transplantation is the last therapeutic option for end-stage lung disease in these patients. Organ transplantation initially requires induction immunosuppressive therapy, followed by maintenance immunosuppression therapy during the years following surgery. This maintenance therapy should be sufficient to prevent organ rejection yet not so intense as to cause over-immunosuppression. Generally, the approach involves combining immunosuppressants with different mechanisms of action and toxicity, aiming to enhance therapy tolerance while minimizing drug dosages. Maintenance immunosuppression is typically achieved through a triple therapy: a glucocorticoid, a calcineurin inhibitor (cyclosporine, tacrolimus), and a nucleotide synthesis blocking agent (azathioprine, mycophenolate mofetil) depending on the hospital’s protocols. Currently, there is no consensus on the optimal regimen for post-lung-transplant maintenance [2,3].

Mycophenolate mofetil, utilized in approximately 70% of lung transplant recipients [4], is a drug that has shown a favorable risk-to-benefit ratio. However, it is subject to extensive inter- and intra-individual response variability [5]. Part of this variability is attributable to pharmacological interactions, which become more likely with the increasing number of regularly taken drugs, as is often the case in individuals with CF.

Here, we present a case of pharmacological interaction between antibiotic therapy and mycophenolate mofetil in a patient with CF undergoing maintenance immunosuppressive therapy. The patient exhibited a probable pharmacological interaction that led to an underexposure to the immunosuppressant due to the antibiotic therapy administered for an exacerbation of lung disease.

## 2. Case Presentation

A 46-year-old white male with CF, diagnosed at the age of 6 at our hospital, underwent a bilateral lung transplant in 2011 followed by immunosuppressive therapy with everolimus and mycophenolate mofetil, regularly monitored through plasma concentrations. He was admitted to the hospital for a pulmonary infectious exacerbation. The patient also has insulin-dependent diabetes mellitus (HbA1c 6.7%) managed with multi-injection therapy (glargine insulin, lispro insulin) and osteoporosis (Z score −3.0) treated with calcium carbonate and cholecalciferol.

Upon admission, he presented with fever, productive cough with greenish sputum, anosmia, and loss of appetite. Initially (day 00), a home treatment with levofloxacin (500 mg/day) led to defervescence for 24 h. However, the fever resumed, and the patient was subsequently treated with sulfamethoxazole-trimethoprim and cephalexin. Severe epigastralgia, vomiting, and diarrhea then occurred. Pulmonary evaluation showed mild respiratory distress and significant fatigue. Due to poor general conditions and acute respiratory symptoms, hospitalization was necessary.

A COVID-19 nasopharyngeal swab test was negative, but the patient tested positive for the H1N1 2009 influenza virus, and antiviral therapy with oseltamivir was initiated. The blood culture was negative, but *Pseudomonas aeruginosa* mucoid type was isolated from the sputum culture. Blood tests revealed elevated C-reactive protein (365.5 mg/L; normal range < 5 mg/L). Consequently, the patient continued sulfamethoxazole-trimethoprim treatment (until day 09) and started (day 03) a continuous infusion therapy with fosfomycin (16 g/day) and piperacillin/tazobactam (16 g/day). Due to the lack of clinical and laboratory response after 48 h, the initial treatment was suspended, and amikacin (1 g/day intravenously) and meropenem (2 g intravenously three times a day) were started.

With progressive clinical improvement, intravenous therapy was continued until day 20, and then oral therapy with azithromycin (500 mg/day) and cephalexin (2 g three times a day) was administered until 14 days after discharge (day 37).

Throughout the hospitalization, the patient continued his antirejection immunosuppressive therapy (everolimus 1.5 mg twice daily, mycophenolate mofetil 500 mg + 1000 mg/day). Plasma concentrations of immunosuppressive drugs measured before admission were 2.30 mg/L for mycophenolate mofetil on day −68, 2.90 mg/L on day −20, and 2.94 mg/L on day 03, which were consistently above the therapeutic threshold of 1.9 mg/L. For everolimus, the measured values were 2.06 µg/L on day −68, 2.66 µg/L on day −20, and 6.27 µg/L on day 03, which were within the therapeutic range of 3–8 µg/L. However, after one week of antibiotic therapy (initially with levofloxacin, sulfamethoxazole-trimethoprim, and cephalexin; then with fosfomycin and piperacillin/tazobactam; and finally with amikacin and meropenem), the pre-dose plasma level of mycophenolate was 0.54 mg/L, which was significantly below the therapeutic threshold. Initially, poor compliance was hypothesized, but this was quickly ruled out after a conversation with the patient conducted by ward staff. On day 12, serial blood samples were taken to evaluate the AUC_0–12_ for mycophenolate mofetil (t_0h_ 0.25 mg/L, t_1h_ 0.26 mg/L, t_2h_ 0.51 mg/L, t_4h_ 1.54 mg/L, AUC 16.38 mg/L h), which turned out to be profoundly below the effective thresholds of: t_0h_ 1.3 mg/L, AUC_0–12_ 60 mg/L h [6].

With the discontinuation of antibiotic therapy, MPA values were gradually found to increase (1.07 mg/L on day 22, 2.49 mg/L on day 38, 4.13 on day 51, 3.35 mg/L on day 123), finally settling above the efficacy threshold of 1.3 mg/L.

## 3. Discussion

MMF is a prodrug that is rapidly hydrolyzed by plasma esterases to form pharmacologically active mycophenolic acid (MPA). MPA inhibits the activity of inosine monophosphate dehydrogenase, leading to the depletion of guanosine nucleotides in T and B lymphocytes. This inhibition thus impedes the proliferation of T and B cells and the glycosylation and expression of adhesion molecules. MPA is finally metabolized by glucuronidation to 7-O-glucuronide (MPAG), an inactive metabolite that is excreted in the urine and feces. Plasma concentrations of MPA are sustained through efficient enterohepatic recirculation, which accounts for up to 40–60% of its AUC_0–12_. This recirculation is a result of the intestinal excretion of glucuronidated metabolites, which are hydrolyzed by enteric flora, predominantly *Bacteroides*, and subsequently reabsorbed [5,7].

In the presented case, the patient was exposed to antibiotic therapy from day 00 to day 23 with various antimicrobial agents: levofloxacin, sulfamethoxazole/trimethoprim, cephalexin, fosfomycin, piperacillin/tazobactam, amikacin, and meropenem (Figure 1). During hospitalization, plasma MPA levels were measured due to suspected therapeutic inefficacy, and a value below the therapeutic range was detected.

MMF is known for its sometimes-unpredictable kinetics, which is why patients undergoing immunosuppressive therapy undergo periodic therapeutic monitoring [5]. For the patient in this case report, the historical values from MPA therapeutic monitoring were consistently above the efficacy threshold, further confirming patient compliance (Table 1 and Appendix A). This regularity made the measured value on day 12 unexpected and deserving of further consideration. After hospitalization, following a week of antibiotic therapy with fosfomycin and piperacillin/tazobactam, and following three days of therapy with amikacin and meropenem, the measured MPA value was significantly below the therapeutic threshold (1.3 mg/L).

The annotation in the Summary of Product Characteristics (SmPC) of the pharmacological interaction, capable of decreasing the immunosuppressant’s plasma concentration when co-administered with amoxicillin/clavulanic acid, ciprofloxacin and norfloxacin, metronidazole, and rifampicin, initially raised the hypothesis of antibiotic therapy contributing to the lower MPA plasma level.

Although the patient was receiving intravenous piperacillin/tazobactam, the structural and activity spectrum resemblance to amoxicillin/clavulanic acid suggested a similar effect on decreasing MPA plasma levels, possibly extending to the entire penicillin class. Later, amikacin and meropenem were added; these potentially further contributed to the interaction, given their extended spectrum of action against Gram-negative bacteria like *Bacteroides*, which appear to be involved in the deglucuronidation of conjugated MPA metabolites.

The SmPC does not mention aminoglycosides, carbapenems, and fosfomycin, perhaps the most intriguing aspect of this case report, warranting further investigation.

The patient had been on oral sulfamethoxazole/trimethoprim antibiotic therapy for several weeks, but this does not appear to have immediately affected MPA plasma concentrations. However, it cannot be ruled out that prolonged antibiotic therapy played a role, even if marginal, in the observed phenomenon.

Although the SmPC and the literature mainly describe an interaction between MMF and oral antibiotic therapy [8], the further contribution of injectable therapy, particularly with agents active against anaerobic Gram-negative bacteria such as *Bacteroides*, cannot be ruled out.

*Bacteroides* are a genus that is particularly abundant in the intestinal environment, where the metabolism of MPAG occurs, releasing MPA. They are anaerobic, Gram-negative bacteria that are particularly sensitive to carbapenems. In the reported case, meropenem was administered for 12 days (from day 05 to day 16), resulting in MPA plasma levels on day 09 and day 12 well below the threshold of effectiveness, specifically 0.54 mg/L and 0.25 mg/L, respectively. This observation may legitimately raise the hypothesis of the involvement of meropenem and probably the entire class of carbapenems.

During the same period, amikacin was also used. Although it is known that aminoglycosides are not effective against anaerobic bacteria like *Bacteroides*, this could, however, suggest the possibility of other genera, distinct from *Bacteroides*, being significantly involved in the hydrolysis of MPA glucuronidated metabolites.

Overall, the observed low plasma concentrations of MPA following polyantibiotic therapy can likely be attributed to the antibiotics’ impact on the intestinal flora, which compromises the regular regeneration of the active compound from its glucuronidated metabolites.

Precisely defining which antibiotics may be most directly involved and which, on the other hand, may play a marginal or clinically insignificant role, remains an open question that requires further investigation.

## 4. Conclusions

Immunosuppression in transplant patients is an area of pharmacotherapy that requires special attention. This is especially important when patients are exposed to multiple drugs that can cause pharmacological interactions, potentially compromising the effectiveness of adequate immunosuppression.

In particular, patients with CF undergoing lung transplantation are particularly prone to pulmonary infections due to their medical condition. Considering the enterohepatic circulation of MMF mediated by intestinal bacteria, vigilant and routine monitoring of MPA concentrations is necessary both during and immediately following the cessation of antibiotic therapies to avoid insufficient immune suppression.

Monitoring plasma concentrations of MPA, coupled with a thorough analysis of the regular pharmacological regimen, during acute exacerbations is a crucial aspect of patient care which can determine the success or failure of immunosuppressive therapy.

## Figures and Tables

**Figure 1 ijms-25-02358-f001:**
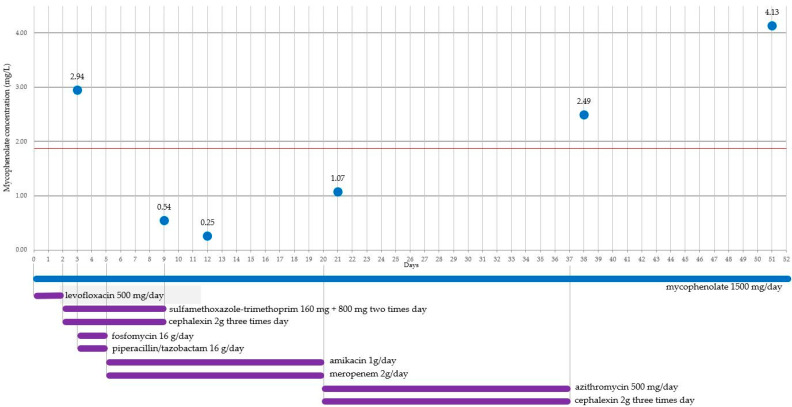
Timeline of measured plasma concentrations of MPA from day 00. The red line represents the minimum efficacy threshold. The administered antibiotic therapy is listed below.

**Table 1 ijms-25-02358-t001:** History of measurements of plasma concentrations of mycophenolic acid (performed by LC-MS/MS method) and everolimus (performed by immunoassay method).

Day	Mycophenolic Acid (MPA)	Everolimus
−88	1.16 mg/L	3.78 µg/L
−83	-	1.88 µg/L
−81	-	2.89 µg/L
−68	2.30 mg/L	2.06 µg/L
−20	2.90 mg/L	2.66 µg/L
3	2.94 mg/L	6.27 µg/L
9	0.54 mg/L	-
12	0.25 mg/L	-
22	1.07 mg/L	3.31 µg/L
38	2.49 mg/L	2.40 µg/L
51	4.13 mg/L	1.98 µg/L
123	3.35 mg/L	2.31 µg/L

## Data Availability

Data are contained within the article and Appendix A.

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
