# Peer review of "Decrease in Mycophenolate Mofetil Plasma Concentration in the Presence of Antibiotics: A Case Report in a Cystic Fibrosis Patient with Lung Transplant"

_ijms, 2024, doi:10.3390/ijms25042358_

Round 1

Reviewer 1 Report

Comments and Suggestions for Authors

The study has a good subject but some points need correction.

1- Abstract: please add 1-2 sentences about the antibacterial interaction with immunosuppressants.

2- Please add the timeline of study starting from day 00.

3- The authors used many antibiotics in high doses. please mention which dose and antibiotics on which days showed major interaction??

4- Is there any mechanism the author thinks worse prognosis for the patient? 

5- Please share flow cytometry results of the WBC range (different WBC groups and populations) before and after the obtained interaction.

6- Please give CRP level and liver enzyme level history after hospitalization.

7- This study is prospective and the patient came to the hospital without planning before; how do you support the study with the grant, also this study needs Ethical permission.

8- Please give antibiotics levels in serum.

Comments on the Quality of English Language

The language needs minor corrections. line 35 caftor, line 37 its to their line 52 an incresing, line 90 and 93 were to was, line 202  scientifc, 203 preventing. please one more time read and correct.

Author Response

Dear Reviewer,

Thank you for taking the time to review our case report.

Below is the response to the points you kindly requested:

1- Thank you for the suggestion. We have added the requested sentence in the abstract.

2- Thank you for the suggestion. Actually, the data at day -20 is redundant because it also appears in the table. We started the time line from day zero. It is also more readable that way.

3- Identifying a specific antibiotic responsible for the interaction can be challenging. We believe the interaction arises from an imbalance in gut flora, which may result from the use of multiple antibiotics. Moreover, pinpointing a particular antibiotic is further complicated by the fact that its impact on gut flora, and consequently its quantitative effect on the reabsorption phenomenon, may take several days to manifest.

4- We don't have elements to identify a further mechanism that could worsen the patient's prognosis, there were no clinical or lab clues suggesting an immunosuppression or a new viral infection. However, we consider a contribution from antibiotics active against gram-negative anaerobic bacteria plausible.

5- We added the required data in a supplementary table.

6- We added the required data in a supplementary table.

7- In accordance with the "General Authorization to Process Personal Data for Scientific Research Purposes" (Authorization no. 9/2014 of the Italian law), which states that retrospective archive studies utilizing ID codes to prevent direct tracing of data back to the data subject do not require ethics approval, this case report, detailing a clinical case observed during routine clinical practice, falls within the scope of the aforementioned authorization. Moreover, according to the Research Institute Policy, informed consent was signed by parents at each visit and admission, in which they agree that “anonymous clinical data may be used for clinical research purposes, epidemiology, study of pathologies and training, with the objective of improving knowledge, care and prevention.”

8- Unfortunately, throughout the course of treatment, plasma levels of antibiotics were not dosed.

Thank you for the feedback about the quality of English language. We have requested a native speaker to review the text to enhance the quality of the manuscript.

Reviewer 2 Report

Comments and Suggestions for Authors

In this paper, the authors present the case of a 46-year-old patient with cystic fibrosis who un-derwent bilateral lung transplantation and was undergoing immunosuppressive therapy.

The authors report that the patient was hospitalized due to an acute pulmonary exacerbation. During the hospitalization, the patient was administered various classes of antibiotics while continuing the standard antirejection regimen of everolimus and mycophenolate. Plasma concentrations of immunosuppressants, measured after antibiotic therapy, revealed significantly lower levels than the therapeutic thresholds. These levels increased after discontinuation of the antimicrobials.

The authors conclude that patients with CF undergoing lung transplantation, who are especially prone to pulmonary infections due to their medical condition, and considering the enterohepatic circulation of mycophenolate mediated by intestinal bacteria, necessitate routine monitoring of mycophenolate concentrations both during and immediately following the cessation of antibiotic therapies that could potentially result in insufficient immunosuppression.

Overall, this is an interesting paper, drawing attention to the interaction of two classes of drugs: antibiotics and immunosuppressants, which is critically important in clinical practice as the use of antibiotics could decrease the levels of immunosuppressants and could favor rejection of the organ transplant.

I commend the authors on this interesting paper.

The knowledge of the potential interaction of antibiotics and immunosuppressants with a decrease in the level of the latter caused by antibiotics has to be known by the medical community.

This interaction of the two classes of drugs is relevant to the transplant community who employs these two classes of drugs.

Author Response

Dear Reviewer,

Thank you sincerely for your encouraging feedback on our manuscript. We truly appreciate your recognition of the importance of the antibiotic-immunosuppressant interaction in transplant patients, and we are committed to providing any additional details or clarifications needed to enhance the quality and comprehensiveness of our article.

Warm regards.

Round 2

Reviewer 1 Report

Comments and Suggestions for Authors

Thank you for the responses and revisions.